# End-of-Life Care in High-Grade Glioma Patients. The Palliative and Supportive Perspective

**DOI:** 10.3390/brainsci8070125

**Published:** 2018-06-30

**Authors:** Giuseppe Roberto Giammalva, Domenico Gerardo Iacopino, Giorgio Azzarello, Claudia Gaggiotti, Francesca Graziano, Carlo Gulì, Maria Angela Pino, Rosario Maugeri

**Affiliations:** 1Neurosurgical Clinic, AOUP “Paolo Giaccone”, PostGraduate Residency Program in Neurologic Surgery, Department of Experimental Biomedicine and Clinical Neurosciences, School of Medicine, University of Palermo, 90127 Palermo, Italy; robertogiammalva@live.it (G.R.G.); Gerardo.iacopino@gmail.com (D.G.I.); Giorgioazzarello1@gmail.com (G.A.); Claudia.gaggiotti92@gmail.com (C.G.); franeurosurgery@libero.it (F.G.); carlogul@yahoo.it (C.G.); mariangelapino@live.it (M.A.P.); 2Department of Experimental Biomedicine and Clinical Neuroscience, University of Palermo, 90133 Palermo, Italy

**Keywords:** astrocytoma, glioblastoma, end of life, supportive care, palliative care, high-grade glioma

## Abstract

High-grade gliomas (HGGs) are the most frequently diagnosed primary brain tumors. Even though it has been demonstrated that combined surgical therapy, chemotherapy, and radiotherapy improve survival, HGGs still harbor a very poor prognosis and limited overall survival. Differently from other types of primary neoplasm, HGG manifests also as a neurological disease. According to this, palliative care of HGG patients represents a peculiar challenge for healthcare providers and caregivers since it has to be directed to both general and neurological cancer symptoms. In this way, the end-of-life (EOL) phase of HGG patients appears to be like a journey through medical issues, progressive neurological deterioration, and psychological, social, and affective concerns. EOL is intended as the time prior to death when symptoms increase and antitumoral therapy is no longer effective. In this phase, palliative care is intended as an integrated support aimed to reduce the symptoms burden and improve the Quality Of Life (QOL). Palliative care is represented by medical, physical, psychological, spiritual, and social interventions which are primarily aimed to sustain patients’ functions during the disease time, while maintaining an acceptable quality of life and ensuring a dignified death. Since HGGs represent also a family concern, due to the profound emotional and relational issues that the progression of the disease poses, palliative care may also relieve the distress of the caregivers and increase the satisfaction of patients’ relatives. We present the results of a literature review addressed to enlighten and classify the best medical, psychological, rehabilitative, and social interventions that are addressed both to patients and to their caregivers, which are currently adopted as palliative care during the EOL phase of HGG patients in order to orientate the best medical practice in HGG management.

## 1. Introduction

Primary brain tumors represent about 2% of the overall cancer diagnoses, and more than 75% of them are high-grade gliomas (HGGs) [1,2,3]. HGG is still nowadays an incurable disease, affecting mostly elderly people. Its incidence is about 5.8 males and 4.1 females per 100,000 inhabitants per year [4]. Stupp protocol with combined surgery, chemotherapy, and radiotherapy still represents the most effective treatment for HGG. Even if the overall survival seems to be related to the surgical resection rate, and the combined therapy has been demonstrated to improve prognosis, HGG still harbors a very poor prognosis, and possibilities to prolong life are limited [2,5,6]. Despite the progresses in treatment, over the last decades survival has not improved, the 5-years survival rate is limited, and the median survival time is about 14 months in younger and healthier patients, whereas elder patients with poor performance status have the worst prognosis [2,3,4,7].

Differently from other types of primary neoplasm, it should be highlighted that HGG is not only a neoplastic disease but, likely because of spinal cord compression by vertebral metastases, it also represents a neurological disease, since it affects the central nervous system [8]. This consideration is to be taken into account when dealing with palliative care and the end-of-life (EOL) phase of HGG patients.

Headache, nausea, vomiting, focal neurological deficits, drowsiness, and prolonged sleep are some of the symptoms of brain neoplasms; they are the targets of HGG palliative care, together with post-surgical morbidities, radiation, chemotherapy, and medical therapy adverse effects [2,9].

Intellectual and cognitive worsening together with mood disorders mark the disease progression and challenge patient’s relatives and caregivers taking care of their loved ones [3].

## 2. End-of-Life and Palliative Care in HGG Patients

The end-of-life (EOL) phase is intended as the time prior to death when symptoms increase and antitumoral therapy is no longer effective [9,10,11]. During this time, patients experience a rapid worsening in their physical, psychological, and social functions [11]. The EOL may range from days to weeks, generally within three months from death [12]. In this phase, medical therapy and cares are aimed to reduce the symptom burden and to maintain the patient’s Quality Of Life (QOL) [9]. Besides physical and cognitive deterioration due to disease progression, the QOL may variously be affected by the lack of symptoms control and the adverse effects of the antitumoral therapy and other medications [9]. During this time, depression, anxiety, and psychological concerns may also add up to the aforementioned symptoms, burdening a patient’s life. Palliative care is intended as an integrated support wich consists of medical, physical, psychological, spiritual, and social interventions which are primarily aimed not to prolong life, but to sustain a patient’s functions during the disease time [9,12]. It has been demonstrated that the early actuation of palliative care can enhance a patient’s QOL, prolong survival, reduce symptom burden, and limit hospitalization [12]. Since over 60% of patients report elevated stress levels [3], it has been demonstrated that not only clinical deterioration should lead to palliative care actuation, but also patient distress and psychological burden [13]. In the EOL phase, caregivers and relatives of HGG patients may be overwhelmed by their role change and the burden of care, experiencing distress, psychological morbidities, and burnout symptoms [14,15]. Palliative care has also been demonstrated to relieve distress in caregivers and increase satisfaction in patient’s relatives [12].

### 2.1. General and Disease-Related EOL Symptoms

HGG patients may experience several general and neurological symptoms during the early and the late EOL phase. Their frequency and severity increase with tumor growth and may be an early sign of progression [16]. Among the HGG disease-specific symptoms, it has been demonstrated that patients may experience more frequently somnolence, motor deficit, headache, dysphasia, cognitive disturbances, seizure, visual disturbances, dysphagia [9,16,17].

Among these, motor deficit, headache, dysphasia, cognitive impairment, seizures, and somnolence are the most frequent symptoms in the early EOL phase, appearing in 31.2–41.9% of patients three months before death [9]. During this time, general EOL symptoms are less frequent: delirium and incontinence are the most prevalent symptoms among HGG patients, with a prevalence of 22.5–23.7% [9]. In the later phase of EOL, general symptoms and disease-specific symptoms are generally more frequent. One week before death, drowsiness and loss of consciousness affect up to 90% of patients [3], dysphasia and cognitive disturbance show a prevalence of 44.7–48.1%, whereas delirium, incontinence, and dyspnoea show an increased prevalence up to 44.4% [9]. In the final days of life, the frequency of dysphagia increases, affecting up to 85% of patients because of motor deficit, muscular weakness, and unconsciousness. In regard to general cancer symptoms, it has been demonstrated that dyspnoea, nausea, depression, and anxiety in HGG patients have a lower prevalence than in the general cancer population [9].

### 2.2. Patient Care and Medical Treatment in EOL

#### 2.2.1. Pain and Headache

Bodily pain shows a lower prevalence than in other types of cancer [9], whereas headache occurs in up to 90% of HGG patients [12]. This may be due to increased intracranial pressure, cerebral oedema, hydrocephalus, or meningeal infiltration [9]. Headache in HGG patients is treated with corticosteroids, in particular dexamethasone. This drug is used in up to 87% of HGG patients, mostly with gastric protection. In case of non-impending cerebral herniation, after one week of treatment, a stable dose of 4 mg/day of dexamethasone is clinically effective and shows the same clinical effects of 16 mg/day [12]. The use of corticosteroid may be enhanced during the EOL phase, but in some patient steroids are withdrawn in the last days of life [9,16]. In this regard, it has been demonstrated that steroids withdrawal does not increase the frequency of symptoms and it may prevent the prolongation of the dying process [9]. In case of corticosteroids failure in lowering the intracranial pressure and relieving headache, ventricular shunting or osmotic therapy with mannitol may ultimately be considered [18]. Non-steroidal anti-inflammatory drugs are also used for the treatment of headache, whereas the use of opioids depends mostly on the care setting and is increased in the last days of life and in case of respiratory discomfort [12]. 

#### 2.2.2. Seizures and Epilepsy

If a patient has never experienced seizures, an antiepileptic prophylaxis is currently recommended only for one week after surgery [19]. However, several patients experience seizures during their EOL phase, and long-term antiepileptic treatment should be started once a patient has seizures [19]. Up to 90% of HGG patients report seizures during the course of their disease, and seizures may increase in severity or develop de novo during the last weeks before death. Moreover, seizures are a frequent reason of rehospitalization [4]. A prevalence of seizure up to 56% has been reported during EOL. The pharmacokinetics of antiepileptic drugs (AED) is a crucial concern when choosing the treatment for a patient with brain cancer, since several AEDs induce cytochrome P-450 (CYP450)-dependent enzymes and influence the metabolism of cytotoxic drugs and even the effectiveness of dexamethasone. New-generation AEDs, such as levetiracetam, do not influence CYP450, so their co-administration with other medications appears to be safer [19]. Among AED, benzodiazepines are widely adopted to treat seizures in HGG patients; however, there is no indication about the preferred form of administration [19]. Intranasal midazolam, rectal diazepam, and buccal clonazepam have shown the same efficacy in HGG-related seizures [18]. Intravenous levetiracetam represents an effective and suitable treatment in patients with dysphagia in hospital care, whereas intramuscular phenobarbital is an effective treatment, easier to manage by caregivers in a home care setting [4,12]. In case of refractory seizures and short life expectancy, palliative sedation with midazolam has been also proposed and may be considered. However, the administration route of AED may differ depending on the care setting [12].

#### 2.2.3. Venous Thromboembolism

Cancer is a known cause of increased venous thromboembolism risk. This risk increases over time, motor deficit may lead to deep vein thrombosis at any time, and the first six months after surgery entail the higher odds of venous thromboembolism. As regards venous thromboembolism prophylaxis, it is usually performed by the use of low-molecular-weight heparin. It should be started within 24 h after surgery in order to avoid the risk of intracranial haemorrhage and it is usually administered for 7–10 days [12].

Common contraindications to venous thromboembolism prophylaxis in HGG patients are recent tumoral bleeding, platelets count lower than 50,000 per mm^3^, and the usual contraindications to heparin administration. In case of venous thromboembolism, there is still no consensus on the duration of the anticoagulation treatment, and low-molecular-weight heparin administration should be planned specifically for each patient [12]. 

#### 2.2.4. Psychological Distress, Depression, Anxiety, and Mood Disturbances

Mood disturbances are common in cancer patients, and depression shows a higher prevalence in HGG patients compared to the general cancer population [16]. It has been reported that depression has a prevalence in up to 28% of HGG patients, but the self-perception of depression is higher, and prevalence may vary depending on the evaluating tools [2,3,12]. Most of HGG patients with depressive symptoms receive proper medications. Moreover, it has been demonstrated that surgery may partially relieve depression [2,3], and this effect may also be correlated to the patient’s pre-operative and post-operative performance status [3]. A previous psychiatric illness and female sex represent risk factors of developing depression [20]. Together with other cancer symptoms, depression severely affects patients’ QOL and it may also determine a faster process of death [3]. Clinical effectiveness of the usually administered medications, such as methylphenidate, oxcarbazepine, bupropion, ginkgo biloba, and donepezil, has not been supported by strong evidence and remains unclear. As regards non-pharmacological therapy, it has been demonstrated, even if without strong evidence, that multimodal psychosocial interventions, massage therapy, acceptance and commitment therapy, and telephonic support may be effective in alleviating depression and anxiety in HGG patients [3,12].

If behavioral disturbance mostly depends on tumor location, depression and anxiety are not correlated to that [3]. Similar to depression, it has been shown that in HGG patients anxiety is correlated to previous psychiatric illness and female sex; moreover, anxiety has also been correlated to lower educational level and uncertainty about treatment [3]. Some evidence supports the use of benzodiazepines as an effective treatment for anxiety, but their use should be cautious because of the high risk of delirium [19].

It has been shown that the highest rate of psychological distress in HGG patient is reported both at the announcement of the initial diagnosis and at the first symptoms relapse [21]. According to the literature, between 28% and 52% of the whole HGG population complains about psychological distress; this may be related to the poor prognosis that patients know to have, to physical and neurocognitive decline, to the sequelae of antitumor therapies they have to endure, and to the uncertainty of the future [21]. Moreover, HGG patients show a higher risk of psychiatric complications compared to the general cancer population, and former psychiatric illness represents a predictor of psychiatric complications during the course of HGG disease [20]. 

During their disease, HGG patients experience a progressive withdrawal from their social lives and usually complain of poor social support during the disease progression. A progressive loss of self, feeling of isolation, loneliness, and vulnerability are common after the diagnosis. The worsening of physical abilities and the progressive loss of independence make patients feel helpless and undignified. The state of uncertainty makes patients feel as they were just waiting and makes them frustrated because they have just cope day by day [22]. During this time, patients’ psychological distress and mood disturbances represent a dynamic psychological process, outweighed by patterns of adjustment, in a difficult adaptation process. According to this, a systematic psychological distress screening should be proposed to HGG patients, since they usually underestimate their distress because of a reduction of insight, and healthcare providers should be properly trained to understand patients’ experience in order to accordingly communicate with empathy [21,23,24]. In this way, tailored support services may relieve patients’ distress, and it has been demonstrated that patients may take advantage of individual and group psychological therapy [23]. In this regard, supportive care should be integrated by psychological services and existential support in the form of a holistic model of care which helps patients to cope with the changes they experience during the disease [2,22]. Among psychological interventions, cognitive behavioral therapy can be helpful in facing psychological concerns such as internalizing problems, anxiety, low mood, somatic complaints, and social and emotional difficulties that HGG patients may encounter [25]. Moreover, an integrated care coordination with an appropriate care plan, psychological interventions, and proper communication between healthcare providers and patients may minimize patients’ sense of uncertainty and may promote a sense of self, thus relieving patients’ psychological distress [2,22,24].

#### 2.2.5. Communication

Patients need precise information about their diagnosis, prognosis, and further therapies, but this may often be overlooked by healthcare professionals, because of insufficient communication skills. The lack of information makes decision-making more difficult both for patients and their caregivers. It has been demonstrated that fear and anxiety in HGG patients appear more related to the perception of an uncertain future rather than to the brain tumor itself, and many patients often want to know everything about their illness [26]. According to this, lower rates of anxiety are reported by patients who are well informed about their illness, their symptoms, the proposed treatment, and possible alternatives. Communication is not only an ethical due but represents an important tool of care since the diagnosis and it may also influence patients’ QOL. Communication should be demanded to properly skilled healthcare professionals, it should be addressed both to patients and their relatives and caregivers, it has to be tailored on the cognitive status and understanding capacity of the patients, and it should compassionately contain some positive messages [2,3].

#### 2.2.6. Cognitive Impairment

In HGG patients, cognitive impairments are more common than in the general cancer population [2] and may result from the tumor itself (in particular from tumor histotype, location, and size) but also from surgery, chemotherapy, and radiotherapy [3]. The worsening of the cognitive status has been related to tumor progression and it may be the prelude of the EOL phase for HGG patients [3]. Cognitive impairments affect more frequently the elderly patients and they increase the odds of depression, fatigue, and a worse QOL [3]. Physiotherapy and occupational therapy have not been related to cognitive improvement. There is no consensus on psychostimulants, which are used with different results [2]. The use of methylphenidate seems to be related to cognitive improvement in HGG patients [27], and donepezil and acetylcholinesterase inhibitors may improve attention, but these results were not confirmed by a randomized placebo-controlled trial [12]. It has been shown that the use of corticosteroids may enhance memory functions, whereas the use of antiepileptic drug may negatively affect them [2]. Interestingly, it has been shown that cognitive rehabilitation and virtual reality training improve the cognitive status in HGG patients, with particular improvements in attention, verbal memory, and visuospatial functions [12]. Younger patients may have the most benefits from these cognitive interventions [12].

#### 2.2.7. Delirium

Delirium is a frequent symptom in the EOL phase of HGG patients; up to 90% of patients experience delirium near death [12]. The causes of delirium are multifactorial, and medications such as benzodiazepines, corticosteroids, and opioids may increase the risk. Despite the high prevalence during the last weeks before death, delirium is often undertreated [9]. Antipsychotics are the drugs of choice; olanzapine, aripiprazole, risperidone, and haloperidol are usually adopted with the same effectiveness [12]. Extrapyramidal effects have been mostly described in association with haloperidol, whereas sedative effects are mostly caused by olanzapine. However, no clinical benefit was attributed to the antipsychotic treatment in a placebo-controlled trial, and the underlying causes should firstly be treated in case of delirium [12,28]. Despite available antipsychotic medications, it seems that benzodiazepines are most often adopted, maybe because of their effect on preventing seizures. In case of palliative sedation because of refractory delirium, benzodiazepines are also adopted as a first choice [4,9,12]. In this case, midazolam at the dose of 5 mg up to four times a day, delivered intravenously, rectally, or submucosally, is a common adopted treatment [4]. 

#### 2.2.8. Fatigue

Fatigue and somnolence are frequent symptoms affecting most of HGG patients since diagnosis [2,12], they are directly related to a worse QOL, and they increase significantly after radiotherapy [2]. Fatigue has been treated with physical exercise, cognitive therapy, and medical therapy, but there is no strong evidence of the effectiveness of these treatments. However, an improvement in fatigue has been reported with the use of methylphenidate, armodafinil, donepezil, and modafinil [12], even if these results have not been confirmed by randomized trials. Patients should avoid extensive naps during daytime to maintain the normal sleep cycle [19]. Resistance training, walking, yoga, and other interventions such as stress management may be useful in alleviating the self-reported rates of fatigue [19].

#### 2.2.9. Nausea and Vomiting

Nausea and vomiting in HGG patients are frequent symptoms of increased intracranial pressure, and their frequency is also related to temozolomide chemotherapy. Since dexamethasone is effective in reducing brain swelling, it is a useful treatment for nausea and vomiting related to increased intracranial pressure [29]. As regards chemotherapy-induced nausea and vomiting, attention should be payed to pharmacological interactions among medications. Metoclopramide, prochlorperazine, and serotonin antagonists are commonly adopted medications in the prevention of nausea and vomiting. However, metoclopramide and prochlorperazine may interact with most antidepressants, and co-administration should be avoided; serotonin antagonists are safer and show less interactions with antidepressants [29]. Granisetron, palonosetron, ondansetron, and dolasetron may also interact with antidepressants and represent a risk factor of serotonine syndrome, whereas granisetron and palonosetron have the lowest interaction rate and the safest profile [29].

#### 2.2.10. Complementary Therapies and Rehabilitation

Negative emotional experiences such as loneliness, sense of loss, fear of death are common in HGG patients [2]. Existential distress often affects patients, and most of them report elevated stress levels [3]. It has been reported that HGG patients often look for complementary therapies in order to reduce their distress and to face their fear through the convincement of having tried everything possible [3]. Up to 41% of HGG patients deploy complementary and alternative medicine with the use of vitamin supplements, homeopathy, self-training, herbs, and faith healing, even if there is no evidence of their effectiveness [3]. Massage therapy has been shown to significantly reduce stress levels in HGG patients, to improve patient’s well-being, and to relieve fears and concerns [3]. Support groups may serve as a way for talking, sharing experiences, and understanding illness, but their role needs to be further examined [2]. Even if there is no evidence in support of them, professional healthcare providers should discuss with patients and their relatives about the use of complementary therapies, their side effects, and their interactions with conventional therapy [12,30].

As regards neurorehabilitation, it has been demonstrated that it may improve functional independence in physical, intellectual, and social functions, while also improving patients’ QOL [3,4,12]. Secondly, rehabilitation may lead to functional gains after radiotherapy, even if it does not improve survival [12]. Moreover, virtual reality and cognitive rehabilitation programs may also improve the cognitive functions, mostly in younger patients [12].

#### 2.2.11. Supportive and End-of-Life Care

Dysphagia is a common symptom in HGG patients; it may be due to the tumor location itself or to the terminal disease, and its frequency increases towards death [4,12]. Swallowing difficulties together with low consciousness and respiratory secretions may avoid oral feeding and may cause death rattle and pulmonary aspiration. Changes in posture may facilitate saliva drainage. There is no evidence that the use of anticholinergic drugs such as hyoscine, atropine, and octreotide may effectively treat the death rattle, but glycopyrronium may reduce noisy breathing [12]. As regards parenteral nutrition and hydration, there is no consensus whether adopting them. Their initiation is related to the EOL decision-making, and their use has to be evaluated singularly for each patient, since ethical concerns on their withholding and their withdrawal vary among different countries [12,18].

#### 2.2.12. Withdrawal of Medications and Terminal Sedation

Withdrawal of medications and supportive care is an ethical issue whose solution varies widely among countries and cultures. Advance care planning may relieve the burden of EOL decision-making both for healthcare professionals and caregivers and ensures that the patients receive the desired care [4,12,31]. During the last weeks of life, terminal sedation may be necessary to obtain a good management of delirium, refractory seizures, and death rattle. However, continuous sedation until death should only be performed in case of refractory symptoms in patients whose life expectancy is no longer than two weeks [18]. Midazolam may be used to obtain a proper sedation, even if withdrawal of steroids may be sufficient to obtain a reduction of wakefulness in the dying phase [4]. 

### 2.3. Organization of Care and Palliative Care Access

Several kinds of palliative care setting are available, and the organization of care varies among countries. Community-based nursing services, social services, rehabilitative therapies, acute hospital services, and hospice units are commonly available options for HGG patients’ care [18]. Even if the place of death may differ, HGG patients mostly prefer to die at home [12]. It has been shown that a good quality of care also depends on the possibility of dying at the preferred place of death, besides being related to receiving satisfactory information and to an effective symptom treatment [11,12]. Moreover, dying at the preferred place of death is considered a predictive factor of dignified death [31]. Factors that may lower the possibility to die at home are admission to the emergency rooms due to inadequate symptoms control or functional deficit, prolonged hospitalization, few home visits by nurses and physicians, social issues, and specific terminal care [12,16,18]. In particular, the lack of symptoms control is a common cause of hospital readmission and transition between different care settings. This results in increasing health care costs and in the worsening of patient’s QOL [18].

### 2.4. Caregivers Burden and Perspectives

HGG represents a challenging disease because of both its difficult management and its features of neurological disease, which represent an important cause of distress in patients’ caregivers [14]. It has been demonstrated that the rate of caregivers’ burdening and distress is higher in the case of primary brain tumors than in the general cancer population [18]. Behavioral changes, physical deterioration, and decline of consciousness force a change in role between patients and their caregivers, since the time of the diagnosis [3,18]. Moreover, lack of communication and feeling unprepared worsen caregivers’ burden [3]. This results in loss of equality in the relationship between a patient and their spouse and in the need of renegotiating the relationship [3]. Several studies have shown a very low QOL, high level of distress, and signs of burnout in caregivers during the EOL phase of their loved one [3,14]. Being young and female are two factors that have been associated with a higher level of distress, and it has also been demonstrated that patients’ well-being, their neurological and psychological status, their health status, and the tumor grade influence caregivers’ well-being [3].

The availability of mobile palliative teams and multidisciplinary support programs may reduce fear and distress in the caregivers of HGG patients [14], and frequent contacts with specialized nurses may alleviate the burden of care [18]. Exercises, massage therapy, meditation, deep soft-belly breathing, coping skills, and progressive muscle relaxation are interventions usually adopted by the caregivers and relatives to relieve their distress [3]. Moreover, a wider involvement in the communication with health care professionals and family consultants is advisable to reduce the caregivers’ stress and burden [3].

### 2.5. Achievement of Palliative Care

When adopted, palliative care avoids unnecessary hospitalization and reduces the health costs. Moreover, home palliative care helps the patients to face their illness and allows them to obtain only the care they want and consider dignified. In this regard, the Advance Care Planning allows the patients to declare and obtain more often the desired interventions, improves the QOL of the patients and their caregivers, and facilitates decision-making during the last phase of patients’ life [4,31,32].

As regards the caregivers’ perspectives, palliative care has positive effects also on caregivers, mostly when assistance is carried out at home [4]. Home palliative care reduces the rate of hospitalization and hospital death and the overall cost of care compared to the hospital stay [4].

## 3. Conclusions

Patients affected by a primary brain tumor have complex needs. HGG patients experience a variety of neurological symptoms together with a progressive physical and cognitive deterioration that require a proper multimodal management. The EOL phase of HGG patients represents a time when no therapy has further effects on disease progression, and only symptomatic and supportive treatment can be adopted. Palliative care in HGG patients during their EOL requires multidisciplinary interventions consisting of medical therapy, rehabilitation, psychological and social support, addressed both to patients and to their caregivers, who experience a high rate of distress and overburdening. Proper and prompt adopted palliative care enhances the QOL of patients and relatives, reduces the health costs, improves patient satisfaction, ensures the fulfilment of patient’s wills, and promotes, in most cases, a dignified death. However, the literature lacks large studies on the EOL phase of HGG patients. Furthermore, clinical, psychological, and social investigations should be performed on palliative care in order to develop new strategies to provide a better EOL care, which seems to be an unheeded part of HGG patients’ management.

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
