# Peer review of "End-of-Life Care in High-Grade Glioma Patients. The Palliative and Supportive Perspective"

_brainsci, 2018, doi:10.3390/brainsci8070125_

Round 1

Reviewer 1 Report

This excellent review gives a good overview on the clinically important topic of end-of-life treatment in patients with malignant glioma. All important issues are discussed well. This work will find the interest of many readers. However, the linguistic style is not sufficient for publication. That makes the review difficult to read and understand. This manuscript needs a thorough text editing prior to publication.

I have listed here some issues:

L15 „Even though it has“ instead of „Even if it has“

L34 „Astrocytoma“ instead of „Astrocitoma“

L82 „early sign of progression“ instead of „early sign of that“

L108 „to an increased frequency of symptoms“

L157 „ginkgo biloba“

Author Response

We performed an English language revision and we also performed the modifications suggested by the Reviewer

Reviewer 2 Report

The author provide a literature review of current end of life care, including palliative and supportive tools, for high-grade glioma patients and subsequently their caregivers. 

The manuscript is comprehensive, providing a breadth of information that is well-organised into sub-headings. 

I would recommend this work for publication provided that the following point are addressed: 

1. There are many grammatical errors (spelling, sentence structure, tenses, word choice). The manuscript should be carefully revised by a native english speaker. Here are a few of the errors I picked up and I would presume that there are more that I've missed: 

- line 93, I think you meant to write "In the final days of life,..."

- line 108 "does not lead to an increase symptom frequency" 

- line 117 "once patient has seizures"

- lines 136/7 "this risk increase over (delete the) time; motor deficit..."

- line 178 "after the diagnosis"

- line 205/6/7 "since diagnosis, ...", "communication should be demanded to".. sentence don't read well

- line 325 referring to general population with a male pronoun (his) is incorrect. Please generalise this to "their"

- line 335 "when (delete prompted) adopted"

- line 354 " Further (insert comma) clinical, psychological..."

2. - In the introduction you state that HGG is itself a neurological disease, great. Then you repeat that statement two paragraphs later in the first sentence of General and disease-related EOL symptoms. I found it a bit repetitive for the reader.

3. There is a lot of information. I felt as though I finished reading the manuscript and had not retained key summary points. To make it easier for the reader, can you create a table organised by your sub-headings (e.g. fatigue, nausea and vomiting, seizures, pain and headache) in column 1 and list current/top strategies for each briefly in a 2nd column (e.g. exercise, dexamethasone)? Maybe a 3rd column with emerging strategies or strategies that not as widely adopted?

Author Response

1-      We performed an English language revision and we also performed the modifications suggested by the Reviewer

2-      We removed the repeated statement, as suggested by the Reviewer

3-      According to the Reviewer suggestion, we created a table where the therapeutic strategies and interventions are briefly summarized.